# A new Majorana platform in an Fe-As bilayer superconductor

Wenyao Liu[1,2,9], Lu Cao [1,2,9], Shiyu Zhu [1,2,9], Lingyuan Kong [1,2,9], Guangwei Wang[3], Michał Papaj[4], Peng Zhang [5], Ya-Bin Liu[6], Hui Chen [1,2], Geng Li [1,2], Fazhi Yang[1,2], Takeshi Kondo[5], Shixuan Du[1,7], Guang-Han Cao [6], Shik Shin [5], Liang Fu[4], Zhiping Yin [3], Hong-Jun Gao [1,2,7✉] & Hong Ding [1,2,7,8✉]

Iron-chalcogenide superconductors have emerged as a promising Majorana platform for topological quantum computation. By combining topological band and superconductivity in a single material, they provide significant advantage to realize isolated Majorana zero modes. However, iron-chalcogenide superconductors, especially Fe(Te,Se), suffer from strong inhomogeneity which may hamper their practical application. In addition, some iron-pnictide superconductors have been demonstrated to have topological surface states, yet no Majorana zero mode has been observed inside their vortices, raising a question of universality about this new Majorana platform. In this work, through angle-resolved photoemission spectroscopy and scanning tunneling microscopy/spectroscopy measurement, we identify Dirac surface states and Majorana zero modes, respectively, for the first time in an iron-pnictide superconductor, $CaKFe_4As_4$. More strikingly, the multiple vortex bound states with integer-quantization sequences can be accurately reproduced by our model calculation, firmly establishing Majorana nature of the zero mode.

[1] Beijing National Laboratory for Condensed Matter Physics and Institute of Physics, Chinese Academy of Sciences, 100190 Beijing, China. [2] School of Physical Sciences, University of Chinese Academy of Sciences, 100190 Beijing, China. [3] Department of Physics and Center for Advanced Quantum Studies, Beijing Normal University, 100875 Beijing, China. [4] Department of Physics, Massachusetts Institute of Technology, Cambridge, MA 02139, USA. [5] Institute for Solid State Physics, University of Tokyo, Kashiwa, Chiba 277-8581, Japan. [6] Department of Physics, Zhejiang University, 310027 Hangzhou, China. [7] CAS Center for Excellence in Topological Quantum Computation, University of Chinese Academy of Sciences, 100190 Beijing, China. [8] Songshan Lake Materials Laboratory, 523808 Dongguan, Guangdong, China. [9]These authors contributed equally: Wenyao Liu, Lu Cao, Shiyu Zhu, Lingyuan Kong. ✉email: hjgao@iphy.ac.cn; dingh@iphy.ac.cn

The iron-chalcogenide superconductor Fe(Te,Se) and its analogous compounds have been found to be able to host isolated Majorana zero modes (MZMs), due to the unique combination of the large superconducting (SC) gap, small Fermi energy, and nontrivial band topology in a single material[1–8]. However, the intrinsic inhomogeneity in Fe(Te,Se) complicates the occurrence of MZMs, causing the topological trivial region in the material[9–11], which brings obstacles for further studies of MZMs and explorations of topological quantum computing. In addition, another family of iron-based superconductors (FeSCs), iron pnictides (Fe-As), which usually have higher values of SC transition temperature ($T_c$) and more abundant crystal structure forms, has long been omitted from the studies of MZMs. While several Fe-As superconductors are predicted to have topological nontrivial Dirac surface states in the previous work, no experimental evidence of MZMs has yet been found[12]. It inspires us to explore a new Majorana-hosting material in the iron pnictide family, which might reveal new physical phenomena and also improve the potential utilization of MZMs.

In this work, we perform a comprehensive theoretical and experimental research on a new type Fe-As superconductor CaKFe$_4$As$_4$ that has a topological band inversion caused by the bilayer band folding. The crystal structure of CaKFe$_4$As$_4$ can be viewed as two different 122-type Fe-As superconductors CaFe$_2$As$_2$ and KFe$_2$As$_2$ inserted into one another (Fig. 1a)[13], where this special structure not only induces high-$T_c$ superconductivity

($T_c = 35$ K) by self-doping effect[14] but also breaks the glide-mirror symmetry along the $c$-axis (Fig. 1a inset). Applying angle-resolved photoemission spectroscopy (ARPES) and the density functional theory (DFT) plus dynamical mean field theory (DMFT) calculation, our investigation indicates that the glide-mirror symmetry breaking together with electron correlations create a topological band inversion in CaKFe$_4$As$_4$. Further measurements confirm the existence of topological Dirac band and its SC state. In addition, by using scanning tunneling microscopy/spectroscopy (STM/S), we observe MZMs within integer-quantization sequence of Caroli–de Gennes–Matricon bound states (CBSs) inside a SC vortex core, which is identified as a topological hallmark of MZMs in the previous studies on Fe(Te,Se)[2,9]. More remarkably, the energy positions and spatial line profiles of multiple bound states can be accurately reproduced by our simple theoretical simulation with all experimental parameters. Our findings demonstrate that CaKFe$_4$As$_4$, with the homogeneous bulk and higher $T_c$, is a new material platform to host and manipulate MZMs.

## Result

**Topological band inversion in the bulk.** The DFT plus DMFT calculations (Figs. 1b–d and 2a) confirm that a topological band inversion can be induced by the glide-mirror symmetry breaking. The glide-mirror symmetry, as seen in CaFe$_2$As$_2$, is broken due to the difference between Ca and K atoms on the opposite sides of

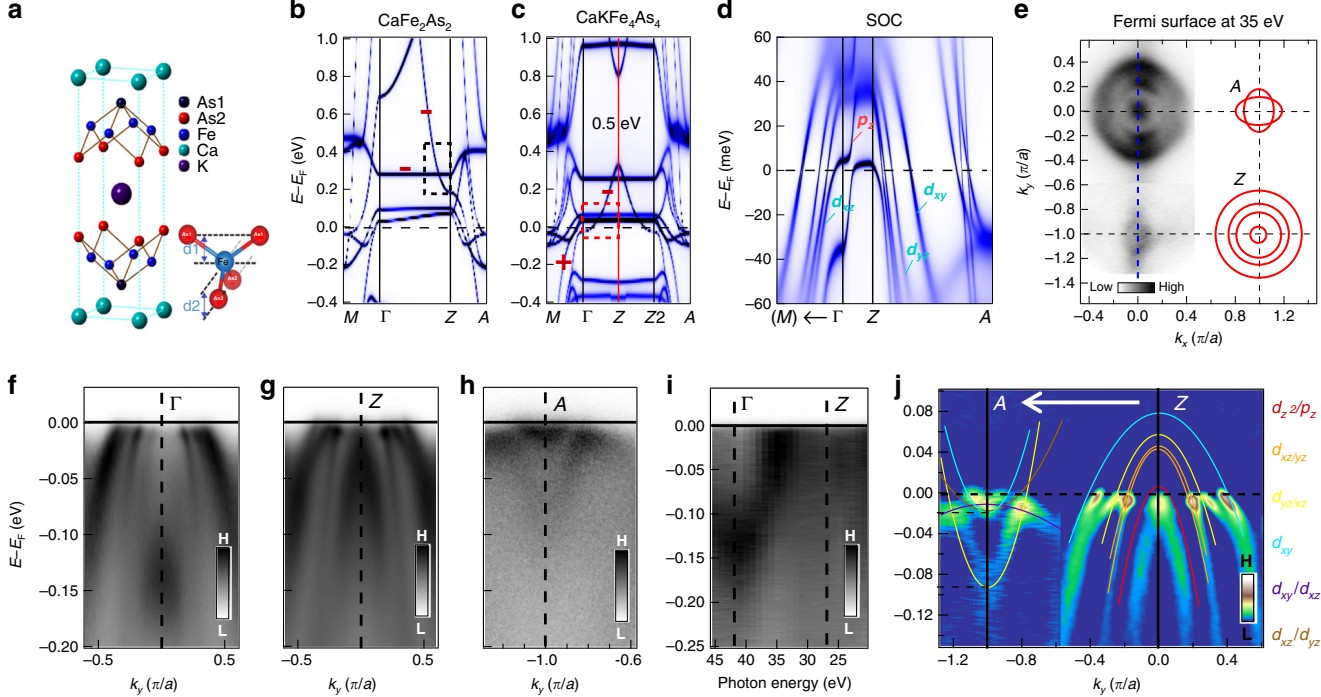

**Fig. 1 Topological band inversion induced by bilayer structure in CaKFe$_4$As$_4$. a** The crystal structure of CaKFe$_4$As$_4$ with the inset of the As-Fe-As tetrahedron. The bond lengths ($d_1$, $d_2$) and angles are different between Fe-As$_1$ and Fe-As$_2$[15]. **b–d** DFT+DMFT calculation results for the band structures of **b** CaFe$_2$As$_2$, **c** CaKFe$_4$As$_4$, and **d** CaKFe$_4$As$_4$ with SOC, respectively. In the **b**, **c**, the red symbols of "+" and "−" represent the band parity, and the red (black) dashed line box marks the position of the topological nontrivial (trivial) band inversion. The glide-mirror symmetry breaking effect in 1144 system is visible by comparing the band structures of CaFe$_2$As$_2$ with CaKFe$_4$As$_4$. In CaKFe$_4$As$_4$, a large hybridization gap (~0.5 eV) is formed between the folded $p_z$ bands, and a band inversion with a SOC gap of 20 to 30 meV is found near $E_F$ at **d**. **e** Fermi surfaces measured by ARPES at a photon energy of 35 eV, which is around the middle point of Γ–Z, note that the '$a$' in scale is 2.73 Å referring to the Fe–Fe distance of CaKFe$_4$As$_4$ in the real space; the red contours on the right are extracted from DFT+DMFT calculation at $Z$ and $A$, which shows good consistence with experimental Fermi surface. **f–h** ARPES spectral intensity plots along the blue dashed line in **e** with $p$-polarized photons at energies of 42 eV (Γ) and 27 eV ($Z$ and $A$). **i** ARPES spectral intensity plot along the Γ–Z direction measured under photon energies from 21 to 45 eV. **j** The momentum distribution curve (MDC) second derivative of the ARPES intensity plot along the $A$–$Z$ direction obtained from **g**, **h**, which enhances the vertical part of the band but suppresses the horizontal part of the band[32], with comparison to the calculated results from **d** (plotted as colored lines).

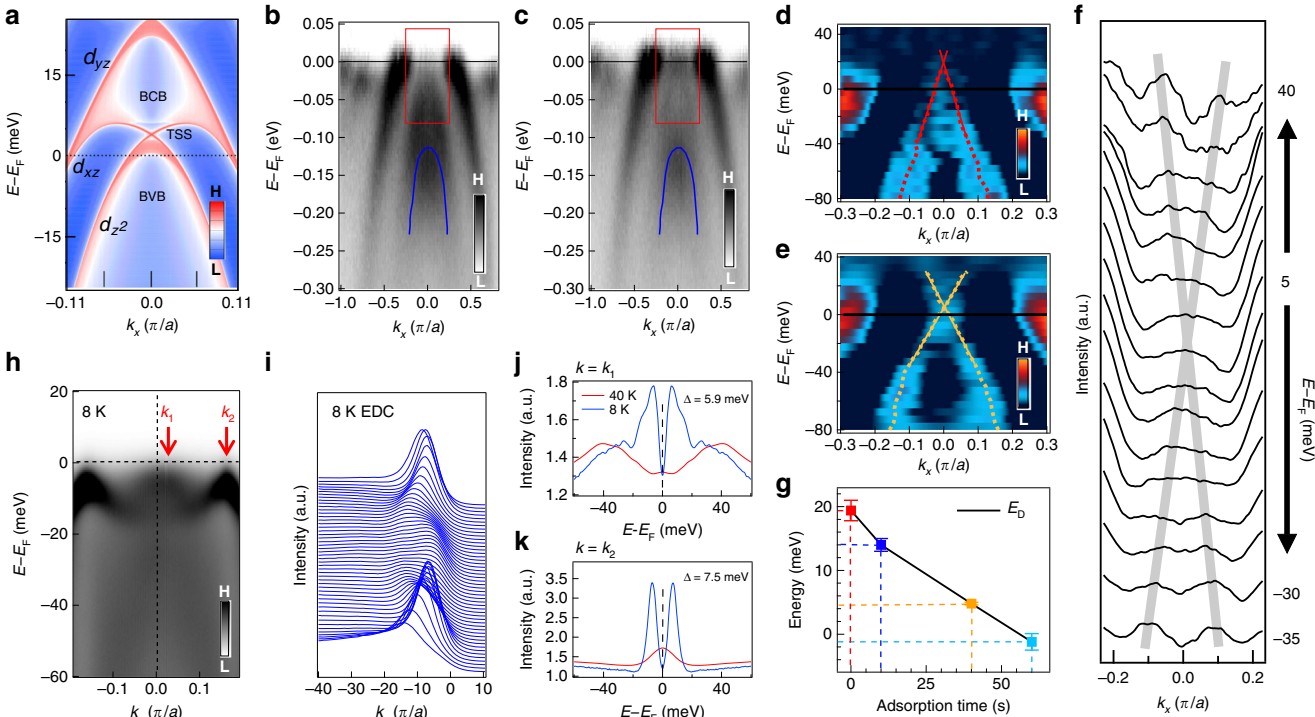

**Fig. 2 Evidence of topological surface state. a** The calculated band structure projected onto the (001) surface. The topological surface state (TSS) along with the bulk valence band (BVB) and the bulk conduction band (BCB) are plotted. **b** ARPES spectral intensity plots along the $\Gamma$–$M$ direction on undoped CaKFe$_4$As$_4$ at the photon energy of 76 eV and the temperature of 25 K; the blue line tracks the $3d_{z^2}/4p_z$ dispersion by Lorentzian fitting on the peaks of MDCs of **b**. **c** Same as **b**, but the sample is under K-surface dosing with current $I = 5.6$ A for 40 s. Both **b**, **c** are divided by the Fermi–Dirac distribution. **d**–**e** The MDC second-derivative plot of **b**, **c** within the red boxes[32]. The red and yellow dashed lines plot the extracted band dispersions, by Lorentzian peaks fitting on the MDCs of the data in **b**, **c**, respectively. A Dirac-cone-like band structure is visible in **e**. The red and yellow solid lines display the linear-fitting results of extracted band dispersions (colored dashed lines). **f** The MDC plot of the data in **c** are normalized by the intensity at $k_x = 0.15$ $\pi/a$ at the energy range from −35 to 40 meV, and the gray line are guides to the eye indicating the positions of peaks. **g** The energies of the Dirac points with the duration of K-adsorption. The error bar is from the uncertainty of the Dirac point, which comes from the different selected ranges of the linear fitting. **h** A laser-based ARPES spectral intensity plot along $\Gamma$–$M$, measured with a $p$-polarized laser (photoenergy 7 eV) at 8 K. **i** EDCs at 8 K. **j**, **k** The symmetrized EDCs at $k_F \sim 0.03$ $\pi/a$ and 0.17 $\pi/a$ (indicated by $k_1$, $k_2$ in **h**) measured at 8 K (blue) and 40 K (red). The superconducting gap values (5.9 meV in **j**, and 7.5 meV in **k**) can be estimated from the two sharp peaks in the symmetrized EDCs. We attribute the gap of 5.9 meV to the SC gap in the topological surface band.

Fe-As layer in CaKFe$_4$As$_4$ (Fig. 1a). Consequently, the Brillouin zone folds along $\Gamma$–$Z$, opening a large hybridization gap (~0.5 eV) at the crossing points of the folded $3d_{z^2}/4p_z$ band, and pushes its lower branch below the $d_{xz/yz}$ band at $\Gamma$, causing a topological band inversion (Supplementary Fig. 1d–f). Finally, the spin–orbital coupling (SOC) opens a topological gap at the band crossing point between $3d_{z^2}/4p_z$ and $3d_{xz/yz}$ bands. We note that our DMFT calculation (Fig. 1b–d) incorporates a mass renormalization of ~5 compared to the simple DFT calculation[15], indicating relatively strong correlations, which can reduce the Fermi energy and the coherence length ($\xi_0$) in this material (details of our calculations are described in Supplementary Materials).

Experimentally, we first performed synchrotron-based ARPES measurements ($h\nu = 21$–45 eV, $T_{exp} = 18$ K). Figure 1e shows the comparison between the measured Fermi surfaces (FSs) at photon energy of 35 eV (left) and the extracted FSs from DFT+DMFT calculation at the $Z/A$ point (right). Self-hole-doping effect is reflected by the large areas of hole-like FSs[16]. Band dispersions near high-symmetry points were also measured (Fig. 1f–h), with a comparison to the DFT+DMFT results (Fig. 1j), showing a fairly good agreement between the two results. Importantly, the innermost hole-like band around $\Gamma$ has a strong $k_z$ dispersion (Fig. 1f, g), so that the hole-like FS pocket around $Z$ sinks well below the Fermi level ($E_F$) at $\Gamma$, which is the consequence of the band inversion between $3d_{z^2}/4p_z$ and $3d_{xz}$ bands. In fact, a clear

band dispersion along $\Gamma$–$Z$ was observed by our ARPES (Fig. 1i), fully consistent with our band calculations (Fig. 1d).

**Evidence of SC Dirac surface states**. We next demonstrate the direct observation of the Dirac surface band around $\Gamma$ ($k_x = k_y = 0$). The calculation of the surface state in CaKFe$_4$As$_4$ (Fig. 2a) shows that the Dirac-cone-type band structure exists inside the SOC gap with its Dirac point above $E_F$, which may obstruct the ARPES technique in observing the Dirac-cone-like feature of the surface state. In order to identify the possible surface states, we implemented the synchrotron-based ARPES measurement on the samples with the in situ potassium (K) surface adsorption that has been proven to induce the effective electron-doping effect on the 122-type FeSC[17]. The experimental photon energy was set to 76 eV, which should be beneficial to detect the electron states of the sample surface when the mean free path of photon-emitted electrons should be small[18,19]. In addition, the band structure (Fig. 2b) in undoped CaKFe$_4$As$_4$ indicates that the selected $k_z$ is around the $\Gamma$ point, while the $3d_{z^2}/4p_z$ (tracked by the blue line) is far below $E_F$. Since there should be a large energy separation between the bulk bands and the surface state near the $\Gamma$ point from the DFT+DMFT calculation, it creates the condition to clearly measure the surface state. Inspiringly, we observed an extra band distinguished from the

$3d_z^2/4p_z$ band, both in the ARPES data (Fig. 2b) and the corresponding second-derivative plot (Fig. 2d), which displays the linear-like dispersion around $E_F$ and cannot be assigned to any bulk band as predicted by our calculation (Fig. 1c, d), suggesting that the newly observed band is the surface state.

To verify the existence of topological surface state, we utilized the potassium (K)-surface adsorption on the sample for 40 s. Note that the surface band dispersion is clearer in the K-surface dosing sample (Fig. 2c) than the undoped sample and displays a good linearity near $E_F$. Moreover, an apparent Dirac-cone-like band can be clearly identified by the second-derivative data (Fig. 2e) and the MDC plot (Fig. 2f) of K-doped CaKFe$_4$As$_4$. We extracted the surface band dispersion under each doping level by a simple Lorentzian fit[20] (the colored dashed lines in Fig. 2d, e). The parameters of the Dirac-cone-like bands are estimated by linearly matching on the tracked dispersions mentioned above (the solid lines in Fig. 2d, e), which suggests the energy of the Dirac point ($E_D$) is ~20 meV and the Fermi momentum ($k_F$) is ~0.025 $\pi/a$ in undoped CaKFe$_4$As$_4$ (red solid line in Fig. 2d). Figure 2g displays the changing of $E_D$ with the K-surface dosing. The systematic shifting of $E_D$ implies the effective doping effect by the in situ K-surface adsorption. The result of the K-doped CaKFe$_4$As$_4$ indicates that the newly observed band in the undoped one (Fig. 2b) is the Dirac surface state, which may induce the topological vortex with MZM.

Then we used the high-resolution laser ARPES ($h\nu \sim 7$ eV) to measure the surface band in the SC state. There is a striking contrast between the sharp spectra in the SC state (Fig. 2h, i) and the broad spectra in the normal state (Supplementary Fig. 7a, b), which is similar to the case in Fe(Te,Se)[21]. From the two Fermi crossings (Figs. 2j, k, $k_1$, $k_2$ in Fig. 2h) near Γ, we can extract the values of two SC gap (5.9 meV for $k_1$ and 7.5 meV for $k_2$).

Since the gap values in most FeSCs, including this material, roughly follow the $\Delta_0\cos(k_x)\cos(k_y)$ formula[20], when measured by ARPES, the SC gap at $k_1$ (closer to Γ) is expected to be larger than the one at $k_2$ if both gaps come from the bulk bands. However, our observation shows that the SC gap in the vicinity of Γ has a smaller size (5.9 < 7.5 meV), which also supports that this SC gap come from the surface band since the smaller SC gap on the topological surface state is likely a proximity SC gap induced by the bulk superconductivity, just as the case in Fe(Te,Se)[1,2]. Combining the band calculation, the band structure data of K-surface dosing and the SC gap measurements, our results strongly support that CaKFe$_4$As$_4$ hosts a topological surface state with its Dirac point just slightly above $E_F$.

**Characterization of sample surface by STM/S.** Encouraged by the promising results from our ARPES measurements and band calculations, we conducted high-resolution (~0.3 meV) STM/S experiments at low temperature ($T_{\exp} = 0.45$ K) to directly search for the signal of MZM inside a vortex core of this superconductor. A typical cleaved surface shows a good atom-resolved topography revealing that the surface is formed by the As lattice[22], with either Ca or K atoms or clusters scattered on top of it (Fig. 3a, b). We chose a flat region with few clusters of Ca or K on which the d$I$/d$V$ spectra are homogenous (Fig. 3c) across a spatial line marked in Fig. 3a. The SC spectra have a main SC gap (Δ) of 5.8 meV, which is likely from the topological surface state with a gap value of 5.9 meV obtained by ARPES, since STM/S is mostly sensitive to the surface. The small bumps at ±3.4 meV are likely coming from the SC gap on the largest hole-like FS since a similar gap value was also observed by ARPES on that FS. The features of SC spectra are similar to the previous results[23,24].

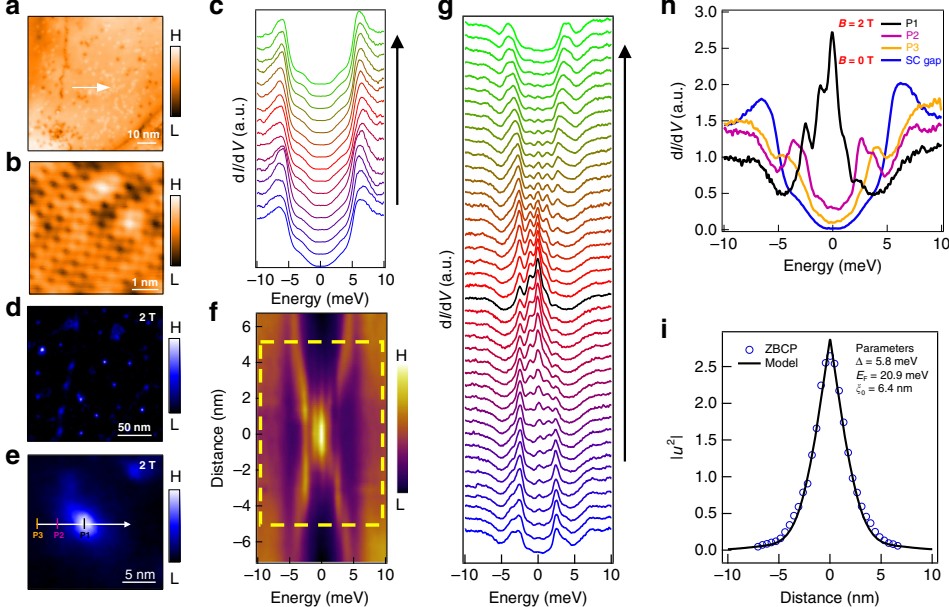

**Fig. 3 MZM in a topological vortex core. a** STM topography of CaKFe$_4$As$_4$ (scanning area: 70 nm × 70 nm). **b** Atom-resolved topography within the area of 5 × 5 nm$^2$. **c** STS spectra taken along the white arrow in **a**, at $T = 0.45$ K and $B = 0$ T. The direction of waterfall-like spectra plot is indicated by the black arrow. **d** Zero-bias conductance map measured at a magnetic field of 2 T in the area as shown in **a**. **e** Zero-bias conductance map (area: 20 nm × 20 nm) around a vortex core at $T = 0.45$ K and $B = 2$ T, the white arrows in **a**, **e** indicate the same position. **f** Intensity plot of d$I$/d$V$ spectra along the white arrow across the vortex in **e**. **g** Waterfall-like d$I$/d$V$ spectra plot within the yellow dashed box in **f**, with the black curve representing the spectrum at the vortex core center, and the black arrow indicates the direction of the plot. **h** Comparison of d$I$/d$V$ spectra at vortex core (P1), middle (P2), edge (P3), and without magnetic field (SC gap). The "SC gap" spectrum is the lowest spectrum in **c** and measured at the same position of P3. All spatial positions are marked in **e**. **i** Comparison between the spatial dependence of the ZBCP peak height with a theoretical calculation of the MZM spatial profile (to exclude the influence of CBSs and background, each d$I$/d$V$ spectrum is fitted by multi-Gaussian peaks to extract the height of ZBCP. An example of a fit is shown in Fig. 4).

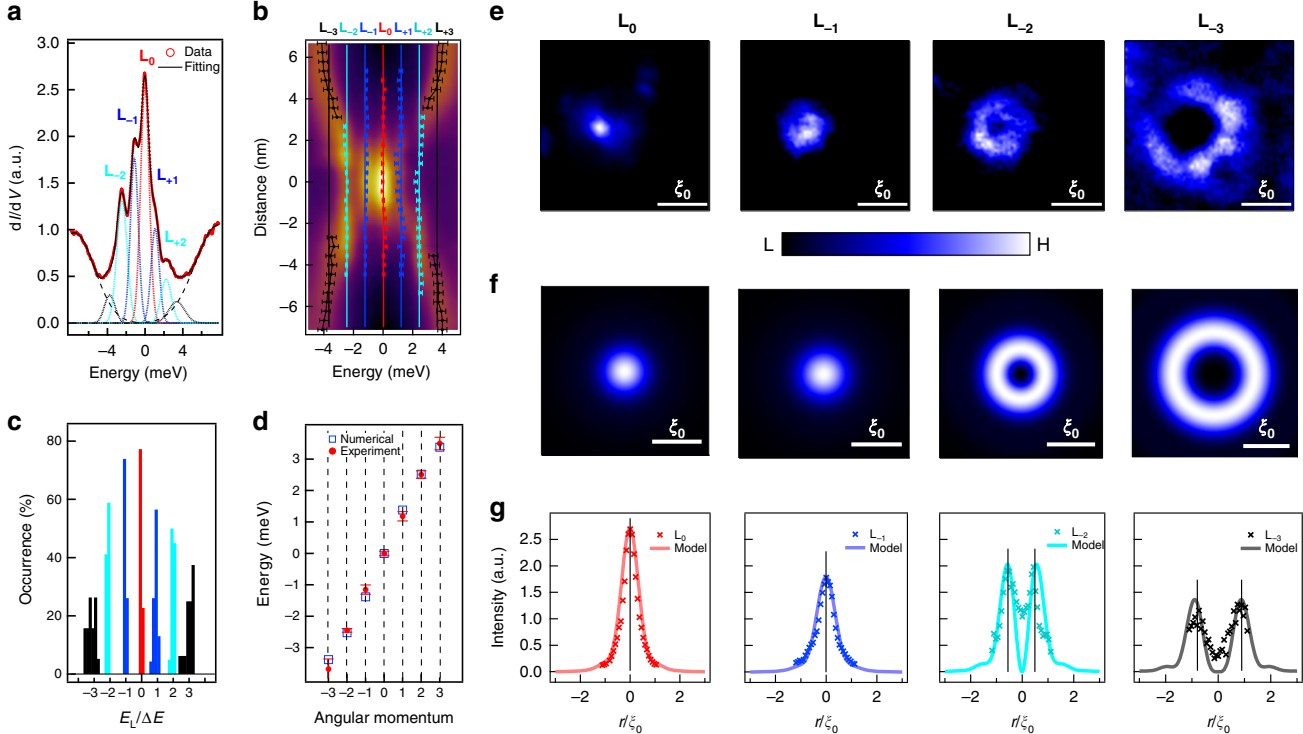

**Fig. 4 Dirac fermion-induced integer-quantized vortex-bound states. a** Multi-Gaussian fit for the d$I$/d$V$ spectrum at the vortex center of Fig. 3f. The red dots are the experimental data, the colored dashed curves are the fitting curves of the vortex-bound states, and the black solid curve is the final fitting result. **b** The line-cut intensity plot same as the Fig. 3f, with the colored marked lines representing the ZBCP and discrete quantized CBSs at different energies marked by $L_0$, $L_{\pm1}$, $L_{\pm2}$, $L_{\pm3}$, which are obtained from the multi-Gaussian fittings for all the spectra in Fig. 3g. The solid lines are calculated using $E_L/\Delta E = n$ ($\Delta E = 1.22$ meV is the average value of energy level spacing), while $n$ is the number of the energy level. **c** A histogram of the energy values of all the observed in-gap states with a sampling width of 50 μeV. The horizontal energy scale is normalized by the first-level spacing. **d** Comparison between the numerically calculated energy eigenvalues of CBSs and experimental values for different angular momenta. **e** Spatial patterns of vortex-bound states at voltage bias equal to 0, −1.2, −2.4, and −3.6 mV, respectively. The size of the area is scaled by the coherence length $\xi_0$ ($\xi_0 \sim 6$ nm for CaKFe$_4$As$_4$). **f** Numerical calculations of the two-dimensional local density of $L_0$ (MZM), $L_{-1}$, $L_{-2}$, and $L_{-3}$, respectively, which are based on the topological vortex core model. **g** Comparison between the wave functions of STM results (crosses) and numerical calculations (lines). The experimental spatial profiles of energy states are extracted along the solid lines in **b**. The intensity of numerical local density of states is rescaled to be comparable to the experimental data.

**MZM within integer-quantized vortex-bound states**. By applying a 2-T magnetic field along the $c$-axis, we clearly observed SC vortices on the surface (Fig. 3d). We focused on one vortex within a $20 \times 20$ nm$^2$ area and measured d$I$/d$V$ spectra along the white arrow across the vortex core (Fig. 3e). A robust zero-bias conductance peak (ZBCP) that does not split or shift as it crosses the vortex can be clearly seen in a line-cut intensity plot (Fig. 3f), and a waterfall-like d$I$/d$V$ spectra plot (Fig. 3g). By utilizing an analytical Majorana wave function derived from Fu–Kane model[2,7], the spatial line profile of the ZBCP is well fitted with the parameters of the Dirac surface state as shown in Fig. 3i ($\Delta = 5.8$ meV, $E_F = 20.9$ meV, $\xi_0 = 6.4$ nm), which are highly consistent with our ARPES and STM data.

Besides the zero mode, there are multiple discrete peaks at finite energies inside the vortex core (Fig. 3h). It is natural to recognize these modes as the quantized CBSs when the electronic temperature ($T_{\mathrm{eff}} \sim 0.69$ K) is much smaller than the quantum limit temperature ($T_{\mathrm{QL}} \sim T_c\Delta/E_F \sim 10$ K)[25,26]. We note that the intensity of CBSs is stronger at the negative energy side, consistent with the scenario that the Dirac point is above $E_F$[13]. As demonstrated recently[9], the CBSs with integer-quantized energy levels, coexisting with a robust MZM, are a hallmark of a Dirac-state-induced vortex-bound states, since the intrinsic $\pi$ phase carried by the spin of surface Dirac fermions lead to an additional half-integer level shift[7,27–30]. To check this behavior,

we used a multi-Gaussian fit to extract the accurate energy positions of discrete bound states inside the vortex (see Supplementary Fig. 2). We resolved seven discrete levels marked by $L_0$, $L_{\pm1}$, $L_{\pm2}$, $L_{\pm3}$ at the energies of 0, ±1.2, ±2.5, and ±3.6 meV, respectively, as shown in Fig. 4a. We displayed the extracted energies of the vortex-bound states at each spatial position onto the line-cut intensity plot (Fig. 4b) and also did a statistics analysis in a histogram plot (Fig. 4c), showing that the discrete vortex-bound states obey an integer quantization with the approximate form of 0: 1: 2: 3, which are derived from the integer angular momenta of vortex-bound states induced by the intrinsic $\pi$ phase of Dirac surface states. Remarkably, we used a model calculation with the same parameter set ($\Delta = 5.8$ meV, $E_F = 20.9$ meV, $\xi_0 = 6.4$ nm) to reproduce the level energies very well (Fig. 4d).

**Characteristic spatial pattern and model calculation.** Due to the high quality of d$I$/d$V$ data measured in a vortex core, we are able to obtain clear observation of spatial patterns for the first four vortex-bound states, i.e., $L_0$, $L_{-1}$, $L_{-2}$, and $L_{-3}$ (Fig. 4e). We observed that both the MZM ($L_0$) and the first-level CBS ($L_{-1}$) have a solid-circle pattern with the maximum of intensity at the vortex center, while the other higher energy CBSs ($L_{-2}$ and $L_{-3}$) show hollow-ring patterns around the vortex center (more results

including CBSs at the positive energy side are displayed in Supplementary Fig. 4). Note that, in an ordinary vortex, only one solid-circle pattern exists, as shown by the theoretical simulation in Supplementary Fig. 4. This phenomenon that both the MZM ($L_0$) and the first-level CBS ($L_{-1}$) possess the maximum-intensity-at-center spatial pattern simultaneously was demonstrated as a fingerprint of surface Dirac fermion-induced vortex-bound states in a topological vortex[9]. In order to simulate the spatial patterns in $CaKFe_4As_4$, we employed a general and direct numerical calculation[7] based on the same set of parameters of the Dirac surface state ($\Delta = 5.8$ meV, $E_F = 20.9$ meV, $\xi_0 = 6.4$ nm) used above. Unlike the simulation in the Fe(Te,Se) case, we reproduced the experimental results with excellent agreement (Fig. 4f, g) without any fine tuning of the gap profile around the vortex, which bodes well with the stoichiometric properties of the $CaKFe_4As_4$ bulk component.

We further note that, while MZM always retains a solid-circle pattern, the same kind of pattern for the first-level CBSs can be only retained on one side, with the other side having a hollow-ring pattern[31]. A positive (negative) Dirac point causes the solid-circle pattern on the negative (positive) side of first-level CBS. Indeed, we observed that the solid-circle pattern appears on the negative ($L_{-1}$) side in $CaKFe_4As_4$, fully consistent with the theoretical simulation based on a Dirac point above $E_F$ (Supplementary Fig. 4). Therefore, all the main features of a topological vortex core, including the spatial line profile of MZM, level energies, and spatial patterns of discrete CBSs, can be fully reproduced by simple model calculations based on a same SC Dirac surface state. This establishes in a convincing manner that this high-$T_c$ Fe-As bilayer superconductor, with glide-mirror symmetry breaking[15], can host isolated MZMs on its surface, offering a new and more practical platform for exploring the properties of MZMs and manipulating them.

## Methods

**Materials and measurement.** Single crystals of $CaKFe_4As_4$ were grown using the self-flux method, and the value of $T_c$ was determined to be 35 K from magnetization and resistivity measurements. Clean surfaces for ARPES measurements were obtained by cleaving samples in situ in an ultrahigh vacuum better than $5 \times 10^{-11}$ Torr. Synchrotron-based ARPES measurements were performed at the "Dreamline" beamline and BL03U of the Shanghai Synchrotron Radiation Facility with Scienta Omicron DA30L analyzers. The energy resolution of "Dreamline" beamline is ~10–15 meV and that of BL03U is ~30 meV. High-resolution laser ARPES measurements were performed at the Institute for Solid State Physics at the University of Tokyo on an ARPES system with a VG-Scienta HR8000 electron analyzer and a vacuum ultraviolet laser of 6.994 eV, with the energy resolution of ~3 meV. The samples used in STM experiments were cleaved in situ ($T_{cleave} = 77$ K) and immediately transferred to an STM head. Experiments were performed in an ultrahigh-vacuum ($1 \times 10^{-11}$ mbar) low-temperature ($T \sim 0.45$ K) STM system of USM-1300-$^3$He with a vector ($9_z - 2_x - 2_y$ Tesla) magnet. Chemically etched tungsten tips were calibrated on Au (111) surface before measurement. Differential conductance (d$I$/d$V$) spectra were acquired by a standard lock-in amplifier at a frequency of 973.0 Hz under the zero-to-peak modulation voltage $V_{mod} = 0.1$ mV. The calibration and configurations of the STM system are shown in Supplementary Note 6. The setpoint of equipment of Figs. 3 and 4 is $V_s = -5$ mV, $I_t = 200$ pA, except Fig. 3a where $V_s = -25$ mV, $I_t = 20$ pA.

## Data availability

The datasets that support the findings of this study are available from the corresponding author upon reasonable request.

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

## Acknowledgements

We thank technical assistance from Y.-B. Huang, S.-Y. Gao, and J.-R. Huang on "Dreamline" synchrotron-based ARPES measurements and D.-W. Shen and Z.-C. Jiang on BL03U synchrotron-based ARPES measurements. This work at IOP is supported by

grants from the National Natural Science Foundation of China (11888101, 61888102, 51991340), Chinese Academy of Sciences (XDB28000000, XDB07000000), the Ministry of Science and Technology of China (2016YFA0202300, 2019YFA0308500, and 2018YFA0305800), and Beijing Municipal Science & Technology Commission (No. Z191100007219012). The band calculations used high-performance computing clusters at BNU in Zhuhai and the National Supercomputer Center in Guangzhou, and Z.Y. is supported by NSFC (11674030), the Funds for the Central Universities (310421113), and the National Key Research and Development Program of China (2016YFA0302300). L.F. is supported by US DOE (DE-SC0019275). Laser ARPES work was supported by the JSPS KAKENHI (Grant Nos. JP18H01165, JP19F19030, and JP19H00651). G.-H.C. is supported by Funds for the Central Universities and the National Key Research and Development Program of China (2019FZA3004, 2017YFA0303002, and 2016YFA0300202).

## Author contributions

H.D. and H.-J.G. designed the experiments. W.L. performed synchrotron-based ARPES measurement; W.L. and P.Z. performed the laser ARPES measurement with assistance from T.K. and S.S.; L.C. and W.L. carried out STM measurements with assistance from S.Z., L.K., H.C., G.L., F.Y., and S.D.; G.W. and Z.Y. performed the DFT+DMFT calculations of surface and bulk band structure; M.P. and L.F. provided theoretical model and simulations of STS results; Y.-B.L. and G.-H.C. synthesized and characterized $CaKFe_4As_4$ single crystals; W.L., L.C., and H.D. analyzed experimental data with inputs from all other authors. W.L. and L.C. plotted figures with inputs from all other authors. All the authors participated in writing the manuscript. H.D. and H.-J.G. supervised the project.

## Competing interests

The authors declare no competing interests.
