## [Peer Review File · Nature Communications]

Reviewers' Comments:

Reviewer #1:

Remarks to the Author:

The work by Liu et al proposes that a new pnictide superconductor can be host of Majorana modes in vortex cores. Authors focus on results obtained only in a few vortices, without investigating the behavior of the lattice nor providing the magnetic field dependence. Authors observe more peaks inside the vortex cores than in previous work, and obtain a few more details using ARPES (without resolving so clearly the superconducting gap though). As for the first point, it is not clear to me how robust these features are, because only one magnetic field is shown and discussed.

Furthermore, there is no comparative discussion of what one should expect in the case of "trivial" superconductivity. My educated guess is that one can reproduce the same results using quasiclassical theory and no Majorana modes. Actually, the observed spatial and energy dependence is more or less the same as predicted for a usual superconductor, with states whose position in space increases in r when you increase the energy. As for the second point, relevant aspects such as coupling with phonons or other excitations that should be in the bandstructure are not discussed. The real aspect where authors could make a punch-line is if they have much better samples than in previous work. But I see no clear-cut evidence for this. And, if that would be the case, I see not such a big difference. Nothing is said about the growth method nor about the sample characterization. Did authors measure lots of samples until they found one with such features? What is the reason for the absence of the observation of such features in previous experiments? Authors should be more careful in discussing this aspect.

Some points to help in future versions of the paper:

- What does it mean "bad metal". I do not recall any evidence for this from other measurements.
- Does the As lattice provide the same pattern as surface reconstruction? Please elaborate. Please provide a Fourier transform of Fig3a. Is the atomic lattice really observed everywhere in the image? The atomic image is too small to be sensibly discussed.
- The sentence "This phenomenon was demonstrated as a fingerprint of surface Dirac fermion induced vortex bound states in a topological vortex" is crucial for the whole paper. It cannot be delegated to another reference. Some explanation is needed. I must say that I do not agree with such an interpretation. Actually, in the calculation provided by the authors, the Majorana mode is not acting. These are just quasiclassical calculations of bound states in a superconductor with a different dispersion.
- As made in previous references, authors should be able to make a direct comparison between ARPES and STM spectra.
- There is no real vortex lattice image (showing hexagonal order) and authors seem to find similar pinning effects as in previous work. It is probably fundamental to see how pinning affects the supposed Majorana mode. Authors should discuss this point.
- At 2 T vortices are clearly very close and there is a strong intervortex interaction. How does this influence the vortex core state?
- The statement that the gap follows a cos behavior is in contradiction with observing cores with a round shape.
- What are the features observed in the upper right part of panel Figure 4 e? Pair breaking or further vortex core features?
- No details about samples are provided. How is their quality, do they observe intergrowth of different phases?
- Predictions by classical Bogoliubov theory in a superconductor with an in plane isotropic gap will provide the same circular patterns. Please compare precisely and show results for calculations made not taking into account Majorana physics.

Reviewer #2:

Remarks to the Author:

This manuscript reports the discovery of Majorana fermions in vortex cores of the topological

superconductor $\text{CaKFe}_4\text{As}_4$. Although Majorana fermions have previously been seen in vortex cores of $\text{Fe}(\text{Se},\text{Te})$ and $(\text{Li},\text{Fe})\text{OHFeSe}$, both of these materials have intrinsic chemical inhomogeneity, which leads to inconsistency in the appearance of the Majoranas. Thus the discovery of Majoranas in the nominally stoichiometric material $\text{CaKFe}_4\text{As}_4$ is important, and worthy of publication in a high-impact journal.

However, I find some of the figures and statements unclear. I have marked detailed comments directly on the PDF file. I hope that the authors will clarify these points, so that I can understand the band structure argument better before making a final recommendation for publication.

The authors should also clarify the measurement parameters that impact the energy resolution of the measurements. For the ARPES images, please take care to submit high-quality vector-format images that preserve the pixelation of the original data. JPEG compression can blur the original pixels and give a dishonest impression of the measurement energy resolution. In particular, I can't tell what was the photon energy resolution. For the STM, how was the electron temperature of the sample quantified? Was the STM bias modulation specified as an rms voltage or 0-peak voltage? (The latter should be clarified in each caption.)

Reviewer #3:

Remarks to the Author:

This paper presents a combined ARPES and STM/STS study of $\text{CaKFe}_4\text{As}_4$. Based on the experimental observation of superconducting Dirac surface state in ARPES and Majorana zero mode inside a vortex core in STM, the authors concluded that this Fe-As based compounds provides a new Majorana platform in Fe-based superconductors.

As the Majorana zero mode was previously found only in iron chalcogenide superconductors $\text{Fe}(\text{Te},\text{Se})$, the results presented in current manuscript represent an important progress in studying the Majorana zero mode in iron-based superconductors. ARPES data for demonstrating the topological surface state and STS spectra for confirming the MZM in the vortex core are complementary in establishing the Majorana nature of the zero mode. The data presented in the manuscript are of high quality, especially the laser photoemission data on the superconducting gap and STS data on the MZM and CBSs. Furthermore, the numeric calculations reproduced the experimental results on the spatial profile of MZM and spatial patterns of CBSs, lending additional support to the claim of MZM in this compound. I would recommend this manuscript for publication in Nature Communications after the authors addressing following comments.

1. In my opinion, the weakest link of the manuscript is the observation of Dirac surface state in ARPES data taken at 200K (Fig. 2c&2d). Comparing with the data taken in $\text{Fe}(\text{Te},\text{Se})$, where the Dirac-cone like dispersion is clearly observed right below the Fermi level, the Dirac surface state is claimed to be above the Fermi level in $\text{CaKFe}_4\text{As}_4$, making it more challenging to observe. Therefore, the authors had to resort to thermal population of states above the Fermi level at very high measurement temperature. Unfortunately, the intrinsically poor statistics and the thermal broadening make the data quality too poor to make a convincing case. In this regard, surface dosing using alkali metal is an alternative approach to access the states above the Fermi level. I would urge the authors to perform this complementary measurements.

2. There appears to be some inconsistency in the superconducting gap measured by ARPES and STS. A superconducting gap of 5.9 meV and 7.5 meV are observed in the laser-ARPES data at two distinct k_F , while the STS spectra show a clear main superconducting gap value of 5.8 meV with small gap edge at ± 3.4 meV that is attributed to the SC gap on the largest hole-like FS. A natural question is why the very pronounced 7.5 meV SC gap in ARPES spectra is completely absent in the STS data. I feel that the authors should discuss this in their manuscript.

3. To me there is still room to improve the data presentation, especially in making better connection between the calculated band structures and band dispersions measured by ARPES in Fig. 1 & 2. The band structure of iron-based superconductors are in general very complicated. It is even worse in case of $\text{CaKFe}_4\text{As}_4$ due to the glide-mirror symmetry breaking. As a result, there are 4-5 hole bands crossing the Fermi level near the zone center. According to the labels in Fig. 1d, the two outermost holes bands have dominant d_{xy} and d_{yz} characters. However, the labeling for the 3 inner hole bands are not so clear, with only d_{xz} and p_z being labeled. In this regard, there are some colored lines on top of the 2nd derivatives plots in Fig. 1j. Clearly the color coding corresponds to the dominant orbital character of each band. A figure legend for those colored lines would be helpful for the readers to compare the band structure calculations to the experimental data. In this regard, there are two nearly degenerate hole bands in the calculated bands highlighted in orange, but there seems to be only one resolved band in the experimental data. What are the orbital characters of these two bands and what is the reason for the discrepancy?

4. There also appears to be clear and important discrepancy between the calculation and experiment in the band top position of the innermost hole band (Fig. 1j). Assuming the band inversion occurs above the Fermi level, we shall expect that the innermost hole band to cross the Fermi level at the Z-point, instead of located below the Fermi level as shown in the data. This discrepancy shall be clarified for a self-consistent picture.

Reviewers' comments:

To Reviewer #1 (Remarks to the Author):

We thank the careful review of Reviewer #1. In the following point-by-point reply, we fully address his questions on material characterization, MBS physics and others issues. In order to follow the format requirements of *Nature Communications*, we write a separated Supplementary Information (SI) to replace the original Method part, and add more results following the suggestions from Reviewer #1 and other reviewers. We have also revised our manuscript accordingly.

The work by Liu et al proposes that a new pnictide superconductor can be host of Majorana modes in vortex cores.

Authors focus on results obtained only in a few vortices, without investigating the behavior of the lattice nor providing the magnetic field dependence. Authors observe more peaks inside the vortex cores than in previous work, and obtain a few more details using ARPES (without resolving so clearly the superconducting gap though).

1. As for the first point, it is not clear to me how robust these features are, because only one magnetic field is shown and discussed.

#Answer:

We performed vortices measurements under two different magnetic fields, *i. e.* 2 T and 4 T, which can be found in Supplementary Fig. S3. We also measured vortices at 5 T which was not shown due to the lower energy resolution of that measurement. Nevertheless, our results on this material, which has a substantial high H_{c2} (> 60 T), show that the Majorana features and other concomitant features induced by different magnetic fields are robust. This result was also quite similar to the one done on Fe(Te,Se) (Ref.2), so we did not emphasize it in this work.

2. Furthermore, there is no comparative discussion of what one should expect in the case of “trivial” superconductivity. My educated guess is that one can reproduce the same results using quasiclassical theory and no Majorana modes. Actually, the observed spatial and energy dependence is more or less the same as predicted for a usual superconductor, with states whose position in space increases in r when you increase the energy.

#Answer:

We are sorry that we are not quite sure what the “quasiclassical theory” refers to. We assume that the reviewer may refer that the vortex states we observed can be simulated in a trivial superconductor by the classical theory where the material is not under the quantum limit. In this case, we agree that a trivial zero bias conductance peak could be observed in the vortex core due to the ultra-small level spacing and limited energy resolution of the equipment. However, considering our experimental electronic temperature $T_{\text{eff}} \sim 0.7$ K, our

experiments should be far below the required quantum limit ($T_{QL} = T_c \Delta / E_F$). The fact that the vortex bound states in our results show up as discrete energy levels with their energies not shifting spatially across a vortex core is a hallmark of the quantum limit, while the quasiclassical theory cannot explain the simultaneous occurrence of the zero-energy mode accompanying with other high energy levels.

3. As for the second point, relevant aspects such as coupling with phonons or other excitations that should be in the band structure are not discussed.

#Answer:

We are not completely sure about the specific meaning of the band structure coupling with excitations like phonons. Nevertheless, we do not think the phonon and other excitations will influence the appearance of Majorana zero modes. The Majorana quasiparticle only cares about low energy physics within the superconducting gap, while the energy scales of phonons and other excitations are usually far beyond the superconducting gap.

4. The real aspect where authors could make a punch-line is if they have much better samples than in previous work. But I see no clear-cut evidence for this. And, if that would be the case, I see not such a big difference. Nothing is said about the growth method nor about the sample characterization. Did authors measure lots of samples until they found one with such features? What is the reason for the absence of the observation of such features in previous experiments? Authors should be more careful in discussing this aspect.

#Answer:

We guess that the feature referred by reviewer #1 is the spatial pattern of vortex bound states. We note while some kind of ring patterns have been observed in our previous work (Ref. 9), they are not as nearly clear as the ones observed in $\text{CaKFe}_4\text{As}_4$. Previous materials like $\text{Fe}(\text{Te},\text{Se})$ or $(\text{Li}, \text{Fe})\text{OHFeSe}$, which require carrier doping to produce their topological band inversion, are inevitably influenced by the inhomogeneity. Such inhomogeneity not only influences the superconductivity but also introduces the topological trivial regions in the sample, which makes the occurrence of MZM more complicated. Therefore, the experimental observation for the good ring pattern in $\text{Fe}(\text{Te},\text{Se})$ or $(\text{Li}, \text{Fe})\text{OHFeSe}$ are more difficult (Ref. 9). $\text{CaKFe}_4\text{As}_4$ is a stoichiometric material, which excludes the problem of doping effect, thus favoring clear observations of such spatial patterns which are in fact strong evidence for existence of the topological Dirac states.

Some points to help in future versions of the paper:

- What does it mean “bad metal”. I do not recall any evidence for this from other measurements.

#Answer:

In the original manuscript, we compared $\text{CaKFe}_4\text{As}_4$ with LiFeAs in order to figure out the commons and differences between these two iron-based superconductors. Since both reviewer #1 and reviewer #2 questioned its necessity, we delete this part in the revised manuscript.

- Does the As lattice provide the same pattern as surface reconstruction? Please elaborate. Please provide a Fourier transform of Fig3a. Is the atomic lattice really observed everywhere in the image? The atomic image is too small to be sensibly discussed.

#Answer:

Due to Ca/K atoms presenting on the surface, it is difficult to obtain a larger image of atomic lattice to do Fourier transformation. The atomic image we shown in Fig. 3b is measured in a relatively clean area without clusters of Ca/K.

- The sentence “This phenomenon was demonstrated as a fingerprint of surface Dirac fermion induced vortex bound states in a topological vortex” is crucial for the whole paper. It cannot be delegated to another reference. Some explanation is needed. I must say that I do not agree with such an interpretation. Actually, in the calculation provided by the authors, the Majorana mode is not acting. These are just quasi-classical calculations of bound states in a superconductor with a different dispersion.

#Answer:

We agree with the suggestion, and modify the manuscript to explain our results more clearly. As for the reviewer’s question, our calculation is based on a widely-accepted theoretical idea (Ref. 7) that the Majorana zero mode can be hosted as a vortex state of a topological surface state with the proximitized s-wave superconductivity. The key point of that proposal is using the spin-momentum-locking characteristic of topological nontrivial surface state to achieve the spinless superconducting state, and creating an effective p-wave superconductivity which is formally equivalent to a spinless p_x+ip_y superconductor except preserving the times reversal invariant. Thus, the magnetic vortex breaks the time reversal symmetry and provides the boundary condition for a Majorana zero mode. In the mathematic formation, the Hamiltonian of the effective p-wave superconductor is simply written as a Dirac surface dispersion combing with an s-wave superconducting pairing, and the zero-energy eigenvalue of magnetic vortex in this system can be rigorously proved to be equivalent to the Majorana zero mode.

- As made in previous references, authors should be able to make a direct comparison between ARPES and STM spectra.

#Answer:

Actually, we have performed such a comparison, and found that the sharp DOS peak around 17 meV below E_F observed in STS spectra is very much consistence with the saddle point observed by ARPES. This data, as shown

below, will be a part of another paper we are currently working on. As for the position of the Dirac surface state, it is more difficult to detect in $\text{CaKFe}_4\text{As}_4$ due to the bilayer splitting which introduces a larger background and totally smears out the dip feature of the Dirac point in STS.

- There is no real vortex lattice image (showing hexagonal order) and authors seem to find similar pinning effects as in previous work. It is probably fundamental to see how pinning affects the supposed Majorana mode. Authors should discuss this point.

#Answer:

One of our manuscript's main points is to capture the Majorana physics in iron-based SC by a simple BdG model for the superconducting Dirac surface state. Therefore, we want to avoid the possible complicated effect including the pinning effect from the localized defect on the surface. The vortices we chose to study were mostly free from surface defect, indicated by the STSs in the same region of the vortex at 0 T (shown in Fig. 3c).

- At 2 T vortices are clearly very close and there is a strong intervortex interaction. How does this influence the vortex core state?

#Answer:

In this material, the coherence length is about 6 nm, which is much smaller than the inter-vortex distance (~ 30nm) under 2 T. Therefore, there should be no impactful Majorana interaction among vortices.

- The statement that the gap follows a cos behavior is in contradiction with observing cores with a round shape.

#Answer:

For a Majorana vortex core, STS measures the local density of states for the zero-energy vortex bound state. Its shape is related to the superconducting symmetry and shapes of the underlying Fermi surfaces.

In $\text{CaKFe}_4\text{As}_4$, the Fermi surfaces are round shape as we observed. One expects that the superconducting pairing

form of $\text{CaKFe}_4\text{As}_4$ is similar with the 122-type Fe-based superconductors, so the superconducting symmetry of $\text{CaKFe}_4\text{As}_4$ is likely to be s_{\pm} wave, and the superconducting gap function should be closely related to $\Delta_0 \cos(k_x) \cos(k_y)$, meaning that the superconducting gap is quite isotropic on each sheet of the Fermi surfaces. Thus, the round shape of vortex cores is a natural result if it is not affected by other factors like impurity.

- What are the features observed in the upper right part of panel Figure 4 e? Pair breaking or further vortex core features?

#Answer:

They are spatial distributions of the vortex state at different levels. We obtained these data by scanning the dI/dV mapping at different voltage biases.

- No details about samples are provided. How is their quality, do they observe intergrowth of different phases?

#Answer:

Our samples are all high-quality single crystal. As suggested, we add the XRD and transport data for sample in the Supplementary Information.

- Predictions by classical Bogoliubov theory in a superconductor with an in plane isotropic gap will provide the same circular patterns. Please compare precisely and show results for calculations made not taking into account Majorana physics.

Answer:

As suggested, we add additional simulation results for the trivial bands in the Supplementary figure 4

The key point for the spatial pattern of Majorana vortex bound state is not solely its circular shape, it should be which level showing up as a solid circle and which level as a hollow circle. The behaviors are different between an ordinary vortex core and a topological vortex core with a Majorana mode.

As for the ordinary vortex state, there is no zero-energy bound state, and all vortex states are circular spatial pattern except either the first or the minus first level state (Ref. 25 and Ref. 28). As a comparison, the topological vortex possesses the zero-energy Majorana bound state, and both the Majorana zero mode and the first (or the minus first) level state have the solid-circle spatial pattern, as shown in the manuscript. These differences are already carefully discussed in the previous paper (Ref. 9). Therefore, the observation of the solid-circle-pattern-spatial-distribution 0 level and -1 level vortex state is a stronger evidence of Majorana zero mode than the observation of a zero bias peak alone.

To Reviewer #2 (Remarks to the Author):

We are grateful for high evaluations and excellent suggestions from the reviewer #2, which are very helpful for us to improve the manuscript. His main concerns are the energy resolution of our experiment and more careful presentation of our manuscript or figures. Below we provide our responses to all the comments and suggestions. We have also revised our paper accordingly.

This manuscript reports the discovery of Majorana fermions in vortex cores of the topological superconductor $\text{CaKFe}_4\text{As}_4$. Although Majorana fermions have previously been seen in vortex cores of $\text{Fe}(\text{Se},\text{Te})$ and $(\text{Li},\text{Fe})\text{OHFeSe}$, both of these materials have intrinsic chemical inhomogeneity, which leads to inconsistency in the appearance of the Majoranas. Thus, the discovery of Majoranas in the nominally stoichiometric material $\text{CaKFe}_4\text{As}_4$ is important, and worthy of publication in a high-impact journal.

However, I find some of the figures and statements unclear. I have marked detailed comments directly on the PDF file. I hope that the authors will clarify these points, so that I can understand the band structure argument better before making a final recommendation for publication.

1. I find it distracting to bring up LiFeAs here, when your paper is about $\text{CaKFe}_4\text{As}_4$. I don't know why you say that iron pnictide has been omitted from the studies of MZMs. Your Ref 12 begins with a discussion of topological states in the iron pnictide $\text{Li}(\text{Fe},\text{Co})\text{As}$.

#Answer:

We agree to focus on $\text{CaKFe}_4\text{As}_4$, and we delete the part of LiFeAs as suggested.

2. If you mention Fig 1e at this point in the sentence, it sounds like it's a calculated figure. But I think it's actually an experimental figure?

#Answer:

Yes, it is an experimental figure. Here we want to compare it with the band structure from DFT+DMFT calculation. We explain our comparison more clearly in the revised manuscript.

3. Can you test the depth-dependence of the states you are seeing, e.g. by photon energy or incident angle of the photon beam?

#Answer:

In principle, it is useful to probe the surface state more directly. However, the response of changing photon energy and incident angle is complicated in practice. The intensity of signal in a multi-band material is

simultaneously affected by the penetrated depth and corresponding cross-section. Similarly, in the previous work of Fe(Te,Se) (Ref. 1), the most suitable photon energy to observe the surface states is close to 7 eV. Moreover, the energy resolution is an important factor to observe topological surface states which usually favors the low photon energy.

4. Is this the macroscopic temperature of the STM (e.g. via a thermometer placed somewhere near the sample) or is this the electron temperature (e.g. measured by the sharpness of coherence peaks of a known superconductor such as aluminum?)

#Answer:

In the manuscript, all experimental temperatures we show are the macroscopic temperature near the sample itself. As a response we put the a Nb-measurement calibration for STM to SI to show the system resolution.

5. Unfortunately, I can't fold the band structure in my head without some more guidance. I think it would be helpful to show a cartoon BZ for the unfolded and folded band structure. It might also be useful to show the folded bands in (c) in a different color, so it's easier to understand the progression from (b) to (c).

#Answer:

We agree, and add the following schematic to explain the band folding in SI. The left panel displays the original band structure (colored as blue) along k_z axis with glide mirror symmetry (GMS) preserving. There is no band inversion between p_z and d_{xz} bands initially. The middle panel shows the folding band (colored as red) after GMS breaking. In this panel, the “Z2” refers to the Z point in the original Brillouin zone (BZ) and “Z” refers to the new Z point in the folding BZ. The right panel shows the folding band structure with the hybridization gap at the edge of the folding BZ. Such the large gap pushes the p_z band below the d_{xz} , thus causing the topological band inversion.

What is the black dashed box in (b)?

What is the red dashed box in (c)?

#Answer:

We highlight the band inversion point in our calculation of CaFe_2As_2 and $\text{CaKFe}_4\text{As}_4$ by using these dashed boxes. The black box indicates the trivial band inversion point in CaFe_2As_2 , and the red box indicates the topological nontrivial band inversion point in $\text{CaKFe}_4\text{As}_4$.

6. I can't understand how (d) relates to (c). The span of (d) is larger than the red box in (c).

#Answer:

The red dash box in (c) indicates the topological band inversion point in the $\text{CaKFe}_4\text{As}_4$ rather than the region of Fig. 1(d) as we explained above. In order to clarify the band folding in $\text{CaKFe}_4\text{As}_4$, the calculated results of CaFe_2As_2 (Fig. 1(b)) and $\text{CaKFe}_4\text{As}_4$ (Fig. 1(c)) are displayed in the same momentum range that G-Z in Fig. 1(b) is same as G-Z2 in Fig. 1(c). In addition, we do not consider the spin orbital coupling (SOC) effect in Fig. 1(b) and Fig. 1(c). Fig. 1 (d) displays only the first Brillouin zone calculation result with the SOC effect in $\text{CaKFe}_4\text{As}_4$, which is different with Fig. 1(c).

7. It's not clear where the red labels pz, dxz, etc. refer to, because there are so many nearby bands. Can you colorcode the bands by their orbital character?

#Answer:

We will modify Fig. 1(d) as suggested.

8. I'm really confused by the axes in Fig 1(f-h). I thought these were supposed to be along the blue line in (e) but that should be the ky axis, whereas these are labeled kx.

#Answer:

Yes. The k_x and k_y values are same in $\text{CaKFe}_4\text{As}_4$ due to its C_4 symmetry. We correct figures as suggested to avoid confusion.

9. What is the actual photon energy resolution in Fig 1(i)? It looks like this image has been run through some kind of Microsoft product that blurs the individual pixels (compresses the image) so I can't tell where one photon energy ends and the next one begins. Please use vector format graphics so you can be honest about your pixelation and energy resolution!

#Answer:

The range of photo energy in Fig. 1(i) is 25-45 eV with the energy resolution being about 18.3-19.7 meV. As suggested, we upload the vector format graphics of Fig. 1(i) again although we already use vector format graphics in our manuscript.

10. I am having a hard time relating this theory plot to the data plots, because the x and y axes is different. Could the authors please use numbers on both axes of (a), or superimpose a rectangle on (a) so we can see the relative k and E-range over which the data plots were acquired?

#Answer:

We add the numerical axis of Fig. 2 (a) as suggested. However, we want to caution that in such the strongly correlated and multi-band material, it is very difficult to obtain a precise agreement between band calculation and experiments.

12. The authors should also clarify the measurement parameters that impact the energy resolution of the measurements. For the ARPES images, please take care to submit high-quality vector-format images that preserve the pixelation of the original data. JPEG compression can blur the original pixels and give a dishonest impression of the measurement energy resolution. In particular, I can't tell what was the photon energy resolution. For the STM, how was the electron temperature of the sample quantified? Was the STM bias modulation specified as an rms voltage or 0-peak voltage? (The latter should be clarified in each caption.)

#Answer:

As suggested, we update the formation of figures of our ARPES data and add the STM calibration part in SI. We also clarify the STM bias modulation in the Method.

In our experiment, the STM bias modulation is specified as zero-to-peak with $V_{modulation} = 0.1$ mV.

To Reviewer #3 (Remarks to the Author):

We sincerely appreciate the thoroughly review and high evaluation of Reviewer #3 on our work. His main questions are the suggestion of surface dosing experiment and the comparisons of DFT+DMFT with ARPES data and STM with ARPES data. We answer those questions in the following point-by-point reply. We have also revised our paper accordingly and added a Supplementary Information (SI) which includes more results and descriptions.

This paper presents a combined ARPES and STM/STS study of $\text{CaKFe}_4\text{As}_4$. Based on the experimental observation of superconducting Dirac surface state in ARPES and Majorana zero mode inside a vortex core in STM, the authors concluded that this Fe-As based compounds provides a new Majorana platform in Fe-based superconductors.

As the Majorana zero mode was previously found only in iron chalcogenide superconductors $\text{Fe}(\text{Te},\text{Se})$, the results presented in current manuscript represent an important progress in studying the Majorana zero mode in iron-based superconductors. ARPES data for demonstrating the topological surface state and STS spectra for confirming the MZM in the vortex core are complementary in establishing the Majorana nature of the zero mode. The data presented in the manuscript are of high quality, especially the laser photoemission data on the superconducting gap and STS data on the MZM and CBSs. Furthermore, the numeric calculations reproduced the experimental results on the spatial profile of MZM and spatial patterns of CBSs, lending additional support to the claim of MZM in this compound. I would recommend this manuscript for publication in Nature Communications after the authors addressing following comments.

1. In my opinion, the weakest link of the manuscript is the observation of Dirac surface state in ARPES data taken at 200K (Fig. 2c&2d). Comparing with the data taken in $\text{Fe}(\text{Te},\text{Se})$, where the Dirac-cone like dispersion is clearly observed right below the Fermi level, the Dirac surface state is claimed to be above the Fermi level in $\text{CaKFe}_4\text{As}_4$, making it more challenging to observe. Therefore, the authors had to resort to thermal population of states above the Fermi level at very high measurement temperature. Unfortunately, the intrinsically poor statistics and the thermal broadening make the data quality too poor to make a convincing case. In this regard, surface dosing using alkali metal is an alternative approach to access the states above the Fermi level. I would urge the authors to perform this complementary measurements.

#Answer:

We appreciate the suggestion from reviewer #3. However, it seems not easy to achieve surface states identification on $\text{CaKFe}_4\text{As}_4$ sample by alkali-metal surface dosing. From our experience, the surface dosing will weaken the intensity of signal and broadening the observed bands (APPLIED PHYSICS LETTERS 105, 172601). We agree that surface dosing may successfully dope the material, however the signal of the Dirac cone most likely fades away. In addition, the alkali ions will damage the sample surface when they are implanted into the surface.

Therefore, we used thermal broadening effect of the high temperature to measure the electron state above E_F as a replacement, and applied the curvature analysis to amplify the feature of band structure above E_F .

2. There appears to be some inconsistency in the superconducting gap measured by ARPES and STS. A superconducting gap of 5.9 meV and 7.5 meV are observed in the laser-ARPES data at two distinct kF, while the STS spectra show a clear main superconducting gap value of 5.8 meV with small gap edge at +/- 3.4 meV that is attributed to the SC gap on the largest hole-like FS. A natural question is why the very pronounced 7.5 meV SC gap in ARPES spectra is completely absent in the STS data. I feel that the authors should discuss this in their manuscript.

#Answer:

Similar with 122-type Fe-based superconductor like $\text{Ba}_{0.6}\text{K}_{0.4}\text{Fe}_2\text{As}_2$, the superconducting gap obeys a three-dimensional distribution on several bands like d_z^2 or $d_{xz/yz}$ in the $\text{CaKFe}_4\text{As}_4$, as a result of strong inter-layer correlation. The gap of $d_{xz/yz}$ is 7.5 meV at a specific k_z , which may change at other k_z . Unlike the ARPES method, which can resolve k_z dispersion, the STS data is an integration of density of state for all k_z and may not have a pronounced 7.5 meV peak which comes from a continuously changing superconducting gap in the reciprocal space. On the contrary, the gap of 5.8 meV is derived of the surface state, which is the SC gap of two-dimensional state. Since the density of state in STS are mostly comes from the surface, it is easy to recognize the SC gap of the surface state by STS.

3. To me there is still room to improve the data presentation, especially in making better connection between the calculated band structures and band dispersions measured by ARPES in Fig. 1 & 2. The band structure of iron-based superconductors are in general very complicated. It is even worse in case of $\text{CaKFe}_4\text{As}_4$ due to the glide-mirror symmetry breaking. As a result, there are 4-5 hole bands crossing the Fermi level near the zone center. According to the labels in Fig. 1d, the two outermost holes bands have dominant dxy and dyz characters. However, the labeling for the 3 inner hold bands are not so clear, with only dxz and pz being labeled. In this regard, there are some colored lines on top of the 2nd derivatives plots in Fig. 1j. Clearly the color coding corresponds to the dominant orbital character of each band. A figure legend for those colored lines would be helpful for the readers to compare the band structure calculations to the experimental data. In this regard, there are two nearly degenerate hole bands in the calculated bands highlighted in orange, but there seems to be only one resolved band in the experimental data. What are the orbital characters of these two bands and what is the reason for the discrepancy?

#Answer:

It is a good comment, and we add the legend in Fig. 1j.

The orbital in orange is $d_{xz/yz}$. Because these two degenerate hole bands are very close to each other and the intrinsic spectrum broadening of the strongly correlated system, it is difficult to separate them in experimental data.

4. There also appears to be clear and important discrepancy between the calculation and experiment in the band top position of the innermost hole band (Fig. 1j). Assuming the band inversion occurs above the Fermi level, we shall expect that the innermost hole band to cross the Fermi level at the Z-point, instead of located below the Fermi level as shown in the data. This discrepancy shall be clarified for a self-consistent picture.

#Answer:

There are two main factors may cause this result.

First, the experimental data in Fig. 1j was measured under 20 K, thus all bands bend near E_F as a result of opening of the superconducting gap. The bending of the innermost hole band is especially large due to the larger superconducting gap of the bulk band. Meanwhile, the k_F value of the innermost hole band is very small which also influences the identification about the location of the band top.

In addition, the k_z broadening of p_z/d_{z2} reduces the weight of intensity on the innermost hole band, when the innermost hole band has the largest k_z broadening.

To identify the band top of the innermost hole band, we extract the band dispersion of Fig. 1j and use a simple parabolic fitting. To exclude the factors mentioned above, we choose the part of band which is far below E_F (red squares). The fitting band is indicated by the red line.

Reviewers' Comments:

Reviewer #1:

Remarks to the Author:

I would like to thank the authors very much for the careful answers. However, to me, this is still not convincing. I would have preferred to see resistivity as sample characterization, not just magnetization (with a doubtful meissner fraction) and x-ray and I share concerns about the interpretation of arpes data. As for the vortex lattice, the argument that the quantum limit is reached is very difficult to maintain if there are in-gap excitations or if the gap is small in a part of the Fermi surface. Furthermore, in the calculations by <https://journals.aps.org/prl/pdf/10.1103/PhysRevLett.80.2921>, there are tons of peaks appearing. As authors show, the ring like structures occur in a trivial vortex core as well. There are measurements at only two magnetic fields and there is no vortex lattice shown, just a few isolated vortices. With all respect to the work and investment of the authors, I need to say that the claims of this paper are, in my view, not supported by the data.

Reviewer #2:

Remarks to the Author:

The authors have addressed all of my major points but one: the photon energy resolution (and resultant kz-resolution) is still not clear.

I would like to know how many photon energies were used in this experiment shown in Fig 1i. If I understand correctly, a single photon energy corresponds to a single momentum between Z and Gamma. The authors now say in the caption that the photon energy ranges from 21-45 eV. Presumably one of these energies corresponds to Z and the other corresponds to Gamma (it might be useful to have another axis label to show this correspondence)? But how many actual momenta do they resolve in between? When I zoom in on this figure, it looks like there are on order 1 pixel per meV, and ~60 pixels spanning from Z to Gamma. This sends a message that the energy resolution is 1 meV, and the momentum resolution is ~1.6% of a BZ. But the authors say in their rebuttal that the energy resolution is ~18 meV. So something is not honest about this Fig 1i. It looks like it has been run through some Microsoft product that compresses the image and blurs neighboring pixels. This is not how data should be represented. It should be in vector format where one measured pixel appears as one rectangle of the correct size, representing the actual measurement resolution.

I have marked some additional points directly on the manuscript – mostly requests for minor wording changes or additional information in figure captions. If the authors address the photon energy resolution, and make the minor requested changes, then I support publication of this manuscript in Nature Communications.

Reviewer #3:

Remarks to the Author:

In the rebuttal letter the authors have provided satisfactory responses to the comments I have raised in my previous report except for the point #1: the lack of convincing experimental evidence for the observation of Dirac surface state in ARPES data. In response to the suggestion of using alkali metal surface dosing as an alternative approach to access the states above the Fermi level, the authors rejected the idea due to the concern over surface degradation upon surface dosing. While I understand the challenge and uncertainty associated with the surface dosing I feel that the authors should at least give a try. There are plenty of successful cases for the alkali metal surface dosing. For example, in a recent ARPES study on topological semimetal GaGeTe (APL Mater. 7, 121106 (2019); <https://doi.org/10.1063/1.5124563>), potassium dosing was proven to be

effective, resulting in a chemical potential shift of nearly half eV with no apparent degradation in the quality of ARPES spectra.

In my opinion, the strength of this paper lies in the combined ARPES and STM/STS study. The most crucial piece of experimental evidence from the ARPES side is the topological surface state shown in Fig. 2, which is unfortunately not very convincing due to the poor data quality using Fermi-Dirac divided 200 K data. The improvement in the ARPES data quality would certainly strengthen the case. Otherwise, the ARPES data alone is insufficient for establishing $\text{CaKFe}_4\text{As}_4$ as a new Majorana platform. In this regard, I also noticed that the Reviewer #1 raised a number of questions on the interpretation of Majorana zero mode inside the vortex core observed in the STM/STS. I am willing to accept the paper based on the complementary information from ARPES and STM/STS, but I would strongly urge the authors to reconsider the surface dosing experiments to improve the credibility of observation of Dirac surface state in ARPES.

To Reviewer #1 (Remarks to the Author):

We thank the careful review of Reviewer #1. We appreciate that the reviewer raised those concerns, *i.e.* the sample quality, the influence of magnetic field and the quantum limit case. Although we think that most of those issues have been addressed in our last response letter and the revised manuscript, we explain the related questions in more details in the following point-by-point reply.

I would like to thank the authors very much for the careful answers. However, to me, this is still not convincing.

Question:

I would have preferred to see resistivity as sample characterization, not just magnetization (with a doubtful meissner fraction) and x-ray and I share concerns about the interpretation of arpes data.

Answer:

There are several previous papers which have characterized the single crystal of CaKFe₄As₄ (Ref.13 and Ref.14 cited in the main text). Following the request of Reviewer #1, we add the resistivity data of our CaKFe₄As₄ crystals in SI. The explanations for our ARPES data are listed in the response for Reviewer #2.

Question:

As for the vortex lattice, the argument that the quantum limit is reached is very difficult to maintain if there are in-gap excitations or if the gap is small in a part of the Fermi surface.

Answer:

So far, vortex bound states under the quantum limit have been clearly observed in the various Fe-based superconductors (e.g. *Phys. Rev. X* **8**, 041056 (2018); *Nat. Comm.* **9**, 970 (2018); *Nat. Mat.* **18**, 811–815 (2019)). As for the concern of Reviewer #1 about this work, we give the following explanations.

Firstly, our experiments have demonstrated that the measured in-gap states are purely vortex bound states from the vortices shown in the main text (Fig. 3c shows the pure superconducting gap spectra without the magnetic field at the same region of the vortex).

Secondly, the previous work (*PRL* **117**, 277001 (2016)) has shown that the superconducting gap of CaKFe₄As₄ obeys the node-less *s*-wave symmetry, and determined the gap sizes of the bulk bands. Based on this work, the smallest Δ^2/E_F value on the Fermi surface is still large enough to reach the quantum limit under our experimental temperature.

Lastly, as we explained in the last response, whether vortex states reach the quantum limit depends on the ratio of the quantization critical temperature ($T_{QL} = T_c \Delta / E_F$) of the sample and the experimental electronic temperature. In CaKFe₄As₄, $T_{QL} \sim 10.5$ K (the superconducting transition temperature $T_c = 35$ K, the size of the superconducting gap on the surface state $\Delta \sim 6$ meV, and the chemical potential $E_F \sim 20$ meV), which is much higher than the

experimental electronic temperature $T_{\text{eff}} = 0.69$ K, promising the appearing condition of the vortex bound states under the quantum limit.

Question:

Furthermore, in the calculations by <https://journals.aps.org/prl/pdf/10.1103/PhysRevLett.80.2921>, there are tons of peaks appearing.

Answer:

The calculation of *Phys. Rev. Lett.* **80**, 2921 is based on the bulk band with a relatively large Fermi energy and a small superconducting gap. The existence of “tons of peaks” is a hallmark of the small quasi-particle gap in this trivial vortex, when the energy spacing of the vortex bound states is scaled with Δ^2/E_F . In addition, the related results about Friedel-like oscillations on the spatial profiles of CBSs are also derived from the calculation based on the trivial case rather than the topological vortex case.

However, in our experiment, the existence of the topological surface states will overwhelm the bulk bands as the emerging vortex bound states, like in Fe(Te,Se) (Ref.2 and Ref.9). The relatively large Δ^2/E_F value of the Dirac surface band enhances the quasi-particle gap and impels the higher levels to be close to the gap edge, causing only a few low levels can be observed inside the superconducting gap.

Question:

As authors show, the ring like structures occur in a trivial vortex core as well.

Answer:

As we explained before, the obvious feature on the spatial patterns of vortex bound states to identify the topological origin is not the ring-like pattern itself. Instead, the differences of the pattern sequences between the topological and trivial vortices are more important, *i.e.* only when the vortex bound states emerge from the topological surface state there are two level states with solid-circle spatial patterns, which has been clearly observed in this work. The supplementary note 5 and the previous work (Ref.9) also give the detailed discussion about this question.

Question:

There are measurements at only two magnetic fields and there is no vortex lattice shown, just a few isolated vortices. With all respect to the work and investment of the authors, I need to say that the claims of this paper are, in my view, not supported by the data.

Answer:

As we explained in the last response, we have measured vortices under different magnetic fields, showing that the Majorana features and other concomitant features are robust under the different magnetic fields. In addition, compared with the ultra-high H_{c2} (> 60 T) of $\text{CaKFe}_4\text{As}_4$, the magnetic field we can achieve in our lab is relatively small (< 8 T), which suggests the relatively low magnetic fields may not influence the MZMs themselves. Moreover, a similar magnetic-field dependent studying of MZMs was already reported in our work on Fe(Te,Se)

(Ref.2). Therefore, we did not emphasize the magnetic-field dependent measurement in this work. As for the question about the vortex lattice, the Majorana vortices we measured were mostly away from the defects on the cleaved surface, which was verified by checking the zero-field STSs (Fig. 3c) at the same region. And the defects trend to prevent Abrikosov vortex lattice forming over a large area under the weak magnetic field applied here.

To Reviewer #2 (Remarks to the Author):

We are grateful for the constructive suggestions from Reviewer #2, which are very helpful for improving our manuscript. Below we provide our responses to all questions and suggestions. We have also revised our manuscript accordingly.

The authors have addressed all of my major points but one: the photon energy resolution (and resultant k_z -resolution) is still not clear.

Question:

I would like to know how many photon energies were used in this experiment shown in Fig 1i. If I understand correctly, a single photon energy corresponds to a single momentum between Z and Gamma. The authors now say in the caption that the photon energy ranges from 21-45 eV. Presumably one of these energies corresponds to Z and the other corresponds to Gamma (it might be useful to have another axis label to show this correspondence)? But how many actual momenta do they resolve in between?

Answer:

The data of the band structure were measured at the photon energies from 21 eV to 45 eV with 1 eV step size. Below we show the raw measurement of k_x - k_z Fermi surface (Left figure below) without the conversion from the photon energy to k_z . Thus, the actual momenta resolved $\sim 5\%$ BZ, and we applied the horizontal linear interpolation function to smooth the raw data as shown in Fig. 1i, which would not change the original band structure displayed below (Right figure). We notice that Reviewer #2 also has some questions about the energy resolution of ARPES method, so we provide a comprehensive explanation in the next answer.

Vector-format Figure of k_z measurement. The Left one displays the Fermi surface along k_z direction, the right one shows the band structure at $k_x = 0$ (cut along red line in the left figure).

Question:

When I zoom in on this figure, it looks like there are on order 1 pixel per meV, and ~ 60 pixels spanning from Z to Gamma. This sends a message that the energy resolution is 1 meV, and the momentum resolution is $\sim 1.6\%$ of a BZ.

But the authors say in their rebuttal that the energy resolution is ~ 18 meV. So something is not honest about this Fig 1i. It looks like it has been run through some Microsoft product that compresses the image and blurs neighboring pixels. This is not how data should be represented. It should be in vector format where one measured pixel appears as one rectangle of the correct size, representing the actual measurement resolution.

Answer:

We notice that Reviewer #2 attempted to obtain the information of the energy resolution by recognizing the size of the pixel. We would like to clarify that the size of pixel represents the setting configuration of our ARPES experimental process rather than the measurement energy resolution.

In an ARPES experiment, the pixel-energy size is related to the scanning step during detecting electrons. In this ARPES measurement, we set the energy scanning step to be 1 meV and used over 300 channels to receive the electrons with the degree range $-15^\circ \sim 15^\circ$ (we recommend the paper *J. Phys.: Condens. Matter* **27** 293203 (2015), which introduces the ARPES method comprehensively). In another word, the size of pixel represents the maximum resolving ability of our detector. The actual resolution of measurement depends on other factors like the thermal broadening effect, the system noise, the linewidth of photon spectrum and the flatness of sample. Generally, we confirm the actual energy resolution in ARPES measurements by the standard sample testing, e.g. measuring the energy broadening of the Fermi-Dirac cutoff of multi-crystal Au. It is natural to infer the energy resolution through the pixel of raw data when the resolving ability of detector is the main factor, however, which is not suitable for our ARPES results here.

Question:

I have marked some additional points directly on the manuscript – mostly requests for minor wording changes or additional information in figure captions. If the authors address the photon energy resolution, and make the minor requested changes, then I support publication of this manuscript in Nature Communications.

1. You didn't refer to Fig 1b-d.
2. If you write "our calculation at the Z point (Fig 1e)", this means that Fig 1e shows your calculation at the Z point. But I think Fig 1e is a measured figure, so your sentence is still misleading. You could say instead: "Fig 1e shows the comparison between the measured Fermi surfaces (FS) at $k_z \sim \pi$ (left) and the calculated FS at the Z point (right)."

Answer:

We are grateful for such careful review. We modified the manuscript as Reviewer #2 suggested.

3. Where can the Fermi energy $E_F=20.9$ meV be seen in Fig 2?

Answer:

It is difficult to infer the energy value of the Dirac point precisely from the APRES data when it is above the Fermi level. Therefore, we extract E_F value by fitting the spatial profile of the zero-bias peak with the theoretical

Majorana wave function derived from Fu-Kane model (Fig 3i). With the new data of potassium-dosing measurement, we observe the Dirac surface band structure, with $E_D \sim 20$ meV in undoped sample, which is in good consistence with the value used before. We modified the manuscript to make this issue clear.

4. Unclear antecedent. Do you mean "The vortex ring phenomenon"?

Answer:

The "fingerprint" here refers to two solid-circle patterns of MZM (L_0) and first-level CBS (L_1) in the topological vortex, which is one of the most distinct differences from the trivial vortex states.

5. It would be helpful to show a 3d BZ cartoon that labels Gamma, Z, A points.

Answer:

It is a good suggestion, and we added a schematic to show the position of each point in the reciprocal space in the Supplementary Figure 1.

6. I still don't see any description in the caption of what the black and red squares mean in b-c. Your caption should also explain the red plus and minus signs (I had to stare at it for a very long time before I realized that the short horizontal red lines were supposed to be minus signs). Your caption must explain all of the features in your figures!

Answer:

We are sorry for the confusion induced by the improper presentation of our figures. We modified all the parts mentioned by Reviewer #2.

7. What does "normalized" mean here? Please describe process in more detail.

Answer:

The "normalized" means the conductance map has been divided by its own average intensity value. Except the absolute intensity value, the normalized map is totally the same as its original one. Usually we use the normalized method to compare different maps with each other, where the unrelated experimental variables like the scanning time should be excluded.

To Reviewer #3 (Remarks to the Author):

In the rebuttal letter the authors have provided satisfactory responses to the comments I have raised in my previous report except for the point #1: the lack of convincing experimental evidence for the observation of Dirac surface state in ARPES data. In response to the suggestion of using alkali metal surface dosing as an alternative approach to access the states above the Fermi level, the authors rejected the idea due to the concern over surface degradation upon surface dosing. While I understand the challenge and uncertainty associated with the surface dosing I feel that the authors should at least give a try. There are plenty of successful cases for the alkali metal surface dosing. For example, in a recent ARPES study on topological semimetal GaGeTe (APL Mater. 7, 121106 (2019); <https://doi.org/10.1063/1.5124563>), potassium dosing was proven to be effective, resulting in a chemical potential shift of nearly half eV with no apparent degradation in the quality of ARPES spectra.

In my opinion, the strength of this paper lies in the combined ARPES and STM/STS study. The most crucial piece of experimental evidence from the ARPES side is the topological surface state shown in Fig. 2, which is unfortunately not very convincing due to the poor data quality using Fermi-Dirac divided 200 K data. The improvement in the ARPES data quality would certainly strengthen the case. Otherwise, the ARPES data alone is insufficient for establishing CaKFe₄As₄ as a new Majorana platform. In this regard, I also noticed that the Reviewer #1 raised a number of questions on the interpretation of Majorana zero mode inside the vortex core observed in the STM/STS. I am willing to accept the paper based on the complementary information from ARPES and STM/STS, but I would strongly urge the authors to reconsider the surface dosing experiments to improve the credibility of observation of Dirac surface state in ARPES.

Answer:

We are grateful for the positive comments from Reviewer #3, and appreciate very much for his deep consideration and helpful suggestions for surface dosing on our experiment, which are very helpful for us to improve the work. Under the recommendation of Reviewer #3, we applied the alkali-metal-surface-adsorption measurement on CaKFe₄As₄ and observed the supportive results that strengthened the evidence for the existence of the Dirac surface state. Below we introduce our new results, and modify the figure (Fig. 2) with K-surface-dosing data in the revised manuscript.

The experimental condition of K-ion dosing is shown as follow: the temperature for both adsorption and measurement are 25 K, the evaporating current of K-ion source is $I = 5.6$ A, and the duration time of adsorption is from 10 seconds to 60 seconds. The K-adsorption effect was quite obvious from the in-situ core level measurement on the sample with different adsorption time. We used 76 eV photons to measure the sample, which should be more sensitive to the surface state. It is obvious from the figures below that in the undoped sample (a) the d_z^2/p_z orbital (guided by the blue line) is much below the Fermi level indicating the k_z is close to the Γ point. Interestingly, there is an extra band dispersion observed clearly above d_z^2/p_z and crosses the Fermi level, which cannot be assigned to any bulk band. More inspiringly, the K dosing made this band clearer (b-c) and a Dirac-cone-like structure appeared on this band reminding the existence of the surface states. We extracted the band dispersion by applying Lorentzian fitting on the peaks of momentum distribution curves, and utilized the linear fitting on the Dirac-cone-type band (for undoped sample, we could not observe the Dirac cone, thus we just

chose the linear part of the band to fit). Our results yield that the $E_D \sim 20$ meV and $k_F \sim 0.025 \pi/a$ in the undoped $\text{CaKFe}_4\text{As}_4$, which is consistent with the value inferred from our previous ARPES and STM data. With K-adsorption time is 40 seconds (e), the Dirac point shifts down to ~ 4 meV above E_F . When the dosing time exceeds 40 s, we find that the band structure of measurement starts to become fuzzy, possibly caused by too much K ions on the sample surface.

Note that, in the most case, the shifting of the chemical potential induced by the alkali-metal-surface dosing on iron-based superconductors is much smaller than the cases on the semiconductors or semimetals, due to the relatively high conductivity of the iron-based materials. In addition, the effect of alkali metal surface dosing on the iron-based superconductors could be very complicated, since the correlation of electrons may be sensitive to the density of electrons and the crystal structure distortion in the materials, *i.e.* the doping effect sometimes is orbital-selective rather than the rigid-band shifting (*Nat. Mat.* **15**, 1233–1236(2016), *Nat. Comm.* **8**, 14988 (2017)).

The core level of K-3p in $\text{CaKFe}_4\text{As}_4$ under the K dosing.

ARPES results of $\text{CaKFe}_4\text{As}_4$ with the K dosing. **a-c**, the data of the band structure under the dosing time of 0 s, 40 s and 60 s, respectively. The blue lines indicate the d_z^2/p_z band dispersion. **d-e**, the second derivative plots of (a)-(c).

Reviewers' Comments:

Reviewer #2:

Remarks to the Author:

The authors have not addressed the points I raised in my round #2 review, so I repeat them here in my round #3 review.

From round #2:

I would like to know how many photon energies were used in this experiment shown in Fig 1i. If I understand correctly, a single photon energy corresponds to a single momentum between Z and Gamma. The authors now say in the caption that the photon energy ranges from 21-45 eV. Presumably one of these energies corresponds to Z and the other corresponds to Gamma (it might be useful to have another axis label to show this correspondence)? But how many actual momenta do they resolve in between? When I zoom in on this figure, it looks like there are on order 1 pixel per meV, and ~60 pixels spanning from Z to Gamma. This sends a message that the energy resolution is 1 meV, and the momentum resolution is ~1.6% of a BZ. But the authors say in their rebuttal that the energy resolution is ~18 meV. So something is not honest about this Fig 1i. It looks like it has been run through some Microsoft product that compresses the image and blurs neighboring pixels. This is not how data should be represented. It should be in vector format where one measured pixel appears as one rectangle of the correct size, representing the actual measurement resolution.

Follow-up in round #3:

The authors now say in the rebuttal that they acquired data with 1 eV energy resolution between 21-45 eV, but they interpolated their figure. However, in Fig 1i, in the manuscript itself, they do not say that the data is interpolated. This is dishonest. The authors should either present the raw data, or they should detail in the caption what manipulations they did on the data as presented. My preference would be to see the raw data (not interpolated!)

Furthermore, the authors have still not made clear how the photon energies correspond to the locations in the Brillouin zone. The Fig 1i caption says "ARPES spectral intensity plot along the Gamma-Z direction measured under photon energies from 21 to 45 eV". But telling the direction does not tell the endpoints. I do not know if 21 eV corresponds to Z or Gamma or some point near Z or Gamma. It would be simplest to just label the photon energies directly on the x-axis of Fig 1a, and also label the Gamma and Z points as applicable. If I zoom in, I should be able to count 25 pixels from left to right on this figure, corresponding to the 25 measured energies from 21 to 45 eV.

In draft #2, line 169, the authors referred to "this phenomena" with unclear antecedent. (Every science paper ever written is about phenomena, so I can't tell without a clarifying adjective which phenomena they are referring to.) In their rebuttal, the authors explained to me which phenomena, but they didn't change the manuscript itself! So on line 201 of the revised manuscript, there is still an unclear reference to "this phenomena". It doesn't help the clarity or viability of this manuscript if the authors just give me, an anonymous reviewer, a private explanation! They have to clarify the manuscript itself!

In draft #2, Fig 3h:

The current legend label "SC gaps" is not informative. I think it should be labeled "B=0". The caption should say where the blue spectrum was acquired (or is it a spatial average?)

In draft #2, Fig 3d caption:

What does "normalized" mean here? Please describe process in more detail.

Follow-up in round #3:

The authors explained "normalized" in their rebuttal, but didn't modify the manuscript. An important goal of this peer-review process is to clarify the manuscript itself for future readers, not just to satisfy some anonymous reviewer once.

Reviewer #3:

Remarks to the Author:

The new surface dosing data in the revised manuscript provides more convincing evidence for the

Dirac-cone-like dispersion in ARPES data. Compared with the previous data recorded at 200K that requires the MDC curvature method to bring out the weak feature, the new data presented in Fig. 2c clearly exhibits a Dirac-cone-like dispersion in the raw data. Furthermore, a systematic shift of the Dirac points as a function of K-dosing lends additional support to the claim of a topological surface state. Therefore, I would recommend this paper for publishing in Nature Communications with no more reservations.

Print Email

Resend E-mail

To Reviewer #2 (Remarks to the Author):

We thank Reviewer #2 for his careful review and suggestions, which are very helpful for improving our manuscript. Below we provide our responses to all his questions and suggestions. We have also revised our manuscript accordingly.

The authors have not addressed the points I raised in my round #2 review, so I repeat them here in my round #3 review.

From round #2:

I would like to know how many photon energies were used in this experiment shown in Fig 1i. If I understand correctly, a single photon energy corresponds to a single momentum between Z and Gamma. The authors now say in the caption that the photon energy ranges from 21-45 eV. Presumably one of these energies corresponds to Z and the other corresponds to Gamma (it might be useful to have another axis label to show this correspondence)? But how many actual momenta do they resolve in between? When I zoom in on this figure, it looks like there are on order 1 pixel per meV, and ~60 pixels spanning from Z to Gamma. This sends a message that the energy resolution is 1 meV, and the momentum resolution is ~1.6% of a BZ. But the authors say in their rebuttal that the energy resolution is ~18 meV. So something is not honest about this Fig 1i. It looks like it has been run through some Microsoft product that compresses the image and blurs neighboring pixels. This is not how data should be represented. It should be in vector format where one measured pixel appears as one rectangle of the correct size, representing the actual measurement resolution.

Question:

Follow-up in round #3:

The authors now say in the rebuttal that they acquired data with 1 eV energy resolution between 21-45 eV, but they interpolated their figure. However, in Fig 1i, in the manuscript itself, they do not say that the data is interpolated. This is dishonest. The authors should either present the raw data, or they should detail in the caption what manipulations they did on the data as presented. My preference would be to see the raw data (not interpolated!)

Furthermore, the authors have still not made clear how the photon energies correspond to the locations in the Brillouin zone. The Fig 1i caption says "ARPES spectral intensity plot along the Gamma-Z direction measured under photon energies from 21 to 45 eV". But telling the direction does not tell the endpoints. I do not know if 21 eV corresponds to Z or Gamma or some point near Z or Gamma. It would be simplest to just label the photon energies directly on the x-axis of Fig 1a, and also label the Gamma and Z points as applicable. If I zoom in, I should be able to count 25 pixels from left to right on this figure, corresponding to the 25 measured energies from 21 to 45 eV.

Answer:

Following Reviewer #2 suggestion, we now replace Fig. 1i with its raw data and add the label of photon energies. We also add the photon energies of data in the caption of the Figs. 1f-h. We apology for the mistake in the figure label of Fig. 1e, and now revise it with a correct one .

Question:

In draft #2, line 169, the authors referred to "this phenomena" with unclear antecedent. (Every science paper ever written is about phenomena, so I can't tell without a clarifying adjective which phenomena they are referring to.) In their rebuttal, the authors explained to me which phenomena, but they didn't change the manuscript itself! So on line 201 of the revised manuscript, there is still an unclear reference to "this phenomena". It doesn't help the clarity or viability of this manuscript if the authors just give me, an anonymous reviewer, a private explanation! They have to clarify the manuscript itself!

Answer:

We sincerely apology for this oversight, and clarify this part in the revised manuscript.

Question:

In draft #2, Fig 3h:

The current legend label "SC gaps" is not informative. I think it should be labeled "B=0".

Answer:

We modify the Fig. 3h mentioned by Reviewer #2. And we also add the black arrows in Figs. 3c&g, which guide the first/last curve and the direction of STS spectra plot.

Question:

The caption should say where the blue spectrum was acquired (or is it a spatial average?)

Answer:

We revise the caption of Fig. 3h suggested by Reviewer #2.

Question:

In draft #2, Fig 3d caption:

What does "normalized" mean here? Please describe process in more detail.

Follow-up in round #3:

The authors explained "normalized" in their rebuttal, but didn't modify the manuscript. An important goal of this peer-review process is to clarify the manuscript itself for future readers, not just to satisfy some anonymous reviewer once.

Answer:

We do not want to complicate the explanation of our data. Since the normalization of data is not necessary here, thus we replace Fig. 3d with the raw data of zero-bias conductance map and delete the "normalized" in the caption.

To Reviewer #3 (Remarks to the Author):

The new surface dosing data in the revised manuscript provides more convincing evidence for the Dirac-cone-like dispersion in ARPES data. Compared with the previous data recorded at 200K that requires the MDC curvature method to bring out the weak feature, the new data presented in Fig. 2c clearly exhibits a Dirac-cone-like dispersion in the raw data. Furthermore, a systematic shift of the Dirac points as a function of K-dosing lends additional support to the claim of a topological surface state. Therefore, I would recommend this paper for publishing in Nature Communications with no more reservations.

Answer:

We appreciate the highly positive comments from Reviewer #3 and his constructive suggestion of surface dosing before.

Reviewers' Comments:

Reviewer #2:

Remarks to the Author:

The authors have satisfied all of my concerns.